# Investigation of positive mental health levels among faculty of health sciences students at a rural university in South Africa

**Rajesh Vikram Vagiri** *, **Phuty Elizabeth Leboho, Lokwene Katlego Desry, Machaka Khutso, Mbedzi Pfunzo**

Department of Pharmacy, Faculty of Health Sciences, University of Limpopo, Mankweng, Limpopo Province, South Africa

* rajesh.vagiri@ul.ac.za

## Abstract

One out of every four people in their lives can be affected by mental health problems that alter their functioning, behaviour, and thinking patterns. In recent years, there has been an increase in mental health disorders among students worldwide. Positive mental health (PMH) has gained relevance in today's fast-paced and demanding world, especially for university students, as it affects their ability to learn, achieve academically, and behave appropriately. This study aimed to investigate the levels of PMH and identify the association between PMH domains and socio-demographic and health-related variables among Faculty of Health Sciences (FHS) students at a rural university in South Africa. A quantitative, descriptive, and cross-sectional survey was conducted. Data was collected using a multidimensional PMH instrument and a socio-demographic and health-related questionnaire, from 354 undergraduate students who are registered for various programmes offered by FHS. The data were analysed using IBM SPSS version 29. Most of the students were black (99.2%, n = 351), single (72%, n = 255), received a study bursary from the government (78.5%, n = 278), hailed from a rural area (77.7%, n = 275) and residing at the university campus (74.6%, n = 246). The total PMH scores of the participants ranged from 4.24 to 4.97 suggesting moderate to higher PMH levels. Significant differences in mean scores were observed in the total PMH and domains of PMH across various socio-demographic and health-related variables. Gender ($p = 0.037$), age ($p = 0.043$) and field of study ($p = 0.016$) showed a significant association with total PMH score. The study's findings highlighted the multi-dimensionality of mental health and justified the importance of evaluating the domains of PMH in university students. The disparities observed across different PMH domains underscore the necessity of embracing innovative approaches to achieve the most effective outcomes to improve mental health and the accurate management of symptoms in students.

**Data Availability Statement:** All relevant data are within the paper and its Supporting Information files.

**Funding:** The authors received no specific funding for this work.

**Competing interests:** The authors have declared that no competing interests exist.

## Introduction

Mental health is, to date, considered a neglected area in developing countries. Mental health problems can affect one out of every four people during their lives, by altering their functioning, behaviour, and thinking patterns [1, 2]. In today's fast-paced society, there is an increasing recognition of the importance of mental health and well-being [3, 4]. Mental health refers to a state of emotional, psychological, and social well-being in which individuals can cope with the stresses of life, work productively, and make meaningful contributions to their communities [5]. Well-being has been proposed as a combination of two traditional approaches: "hedonic"- and "eudemonic" concepts. The hedonic concept of well-being has been defined as "high positive affect, low negative affect and high life satisfaction" which focuses on the feelings of individuals towards life. Whereas the eudemonic concept focuses on functioning in life, which includes the dimensions of self-acceptance, personal growth, autonomy, relationships, and environmental mastery [6].

In a fast-paced and demanding society, positive mental health (PMH) has become a topic of increasing importance, particularly among university students [6, 7]. The concept of PMH goes beyond the absence of mental illness and encompasses the cultivation of emotional resilience, well-being, and overall psychological flourishing [4, 8, 9]. The concept of PMH was further supported by a study conducted by Keyes (2000) who described mental health as a "syndrome of symptoms of positive feelings and positive functioning in life". A diagnosis of the presence of mental health, described as flourishing, and the absence of mental health, characterised as languishing. The study findings reported a risk of a major depressive episode that was two times more likely among languishing than moderately mentally healthy adults and nearly six times greater among languishing than flourishing adults [10].

### Positive psychology in South Africa

The burden of mental disorders in South Africa has significantly increased over the past two decades, contributing substantially to the country's overall disease burden [11]. This rise in mental health issues has led to a growing concern about the state of mental health services in South Africa, with evidence indicating the necessity for population-wide and individual-level interventions to enhance mental health literacy [12]. The Mental Health Policy Framework and Strategic Plan for South Africa (2013–2020) emphasised the importance of primary health care, multisector collaboration, and addressing issues such as stigma and gender in the country's mental health policy framework [13].

The current state of positive psychology in South Africa is influenced by various factors, including the mental health policy framework, the treatment gap for mental disorders, the burden of mental disorders, the state of mental health services, and historical and contextual factors [14]. South Africa's mental health policy framework underscores the need for integrated mental health care within primary health care, presenting both challenges and opportunities for the governance of the health system to support this integration [15]. Additionally, the experiences and perceptions of mental health service provision at primary health centres in South Africa indicate that mental healthcare remains sparse, while the impact of mental and substance use disorders continues to add to the burden of disease in African countries [16].

The stress-generating environment in South Africa has implications for the mental health of its population [17]. Apartheid has had a profound impact on South African psychology, with historical and contextual factors shaping the trajectory of psychological work on health issues [18]. Post-apartheid South Africa continues to grapple with endemic social problems that create conditions of oppression and violence, necessitating new ways of engagement in activism for improved mental health [13]. The challenges faced by South Africa, such as the

HIV epidemic and the barriers to mental healthcare have further underscored the need for a comprehensive understanding of mental health in the country [19, 20].

A study conducted by Coetzee and Viviers (2007) categorised South African research in positive psychology over the last 36 years, providing insights into the diversity of the fields of psychofortology and related disciplines represented by the published articles. It highlighted the lack of a cohesive foundational theoretical framework and the challenge of expanding a classification context to synthesise diverse states, traits, and outcomes for each other [21].

## Differences in positive mental health between the urban and rural residents of South Africa

The socio-economic and cultural divide between urban and rural settings in South Africa significantly impacts human settlement and well-being. The environmental setting plays a major role in what one becomes, partly through determining the quality of education available, possible opportunities for formal employment, and quality of lifestyle [22]. Ataguba et al. (2011) highlighted the intertwining of human settlement with socio-economic factors, influencing living conditions and subjective well-being [22]. Rural areas in South Africa are characterised by poverty and underdevelopment related to the experience of psychological distress and ill health by rural residents. Individuals living in poverty experience lower levels of happiness, indicating the impact of socio-economic conditions on well-being [22, 23]. Mthembu et al. (2017) observed an increase in psychological distress with rural residents, associated with poverty and underdevelopment, leading to psychological distress and ill health among residents [24].

Although literature exists on the impact of urban living on physical health and mental illness, there is very little on psychosocial well-being. A study conducted in the North West Province of South Africa explored the socio-demographic variables and psychosocial well-being of an African group in rural and urban areas using the General Psychological Well-being (GPW) and the Mental Health Continuum (MHC) models. The study findings revealed that urban participants reported higher levels of psychosocial well-being in most facets of individual and social well-being compared to rural residents. The above findings further suggest that the current state of African rural life is detrimental to well-being [23].

## Positive mental health of university students in South Africa

Globally, there has been a rise in mental disorders among students in recent years. According to the WHO World Mental Health International College Student Project conducted at 19 colleges in eight developed and developing countries, that colleges are contending with increased rates of mental health conditions. Students reported a combination of mental disorders such as major depression, mania, anxiety disorders, panic disorder, alcohol use disorder, and substance use disorder that impacted their academic performance and overall well-being [25]. In response to this alarming trend, it is crucial for higher education institutions to actively address the mental health needs of their students by creating a healthier campus environment that supports PMH and well-being [26–28].

Additionally, there is a growing recognition of the need for preventive mental health practices among university students [6, 7]. As students navigate the pressures of academics, social interactions, and personal development, it is crucial to understand and support their PMH [29, 30]. Understanding and promoting PMH among university students is vital to ensuring their holistic development and academic success [31]. Furthermore, a systematic review by Hobbs et al. (2022) highlighted the importance of addressing poor psychological well-being in university students, as it can impair academic performance and increase the likelihood of

dropping out. The study recommended rigorous evaluation of positive psychology and the implementation of interventions to promote students' psychological well-being [32].

In recent years, few studies have been conducted in South Africa to assess mental well-being, focusing on the investigation of the prevalence of mental health symptoms or conditions among university students based in urban areas. Bantjes et al. (2019) conducted a study on the prevalence and socio-demographic correlates of common mental disorders (CMDs) among first-year university students in post-apartheid South Africa [33]. Another study investigated the prevalence and factors associated with mental distress among university students in the Eastern Cape Province, South Africa [34]. Both studies highlighted the need for targeted interventions to address mental health issues and support the mental well-being of university students.

On the other hand, few studies have explored the influence of COVID-19-related experiences on the mental health of South African university students. This research sheds light on the impact of the COVID-19 pandemic on the mental health of university students, providing valuable insights and a deeper understanding of the challenges faced by this population during the pandemic [35, 36].

The above studies investigated the mental well-being of the student population in South Africa and provided comprehensive insights into the mental health challenges faced by South African university students, including the prevalence of mental disorders, the impact of stressful life events, and the influence of the COVID-19 pandemic on mental well-being. However, none of the studies assessed various domains of PMH among university students in South Africa, especially in a rural setting. A PMH for students is very important as they navigate the challenges of higher education and face various stressors such as academic pressure, financial burdens, and social expectations. These stressors can have a significant impact on their overall well-being and academic performance [37–39]. Therefore, this study aimed to investigate the levels of PMH and its association with socio-demographic and health-related variables among FHS students at a rural university in South Africa.

## Methods

### Study design and population

A cross-sectional survey was conducted among Faculty of Health Sciences (FHS) students at a rural university in South Africa. A cross-sectional research design was used because it allows studies to collect data to make inferences about a population of interest at one point in time. This study followed a descriptive and quantitative approach that included FHS undergraduate students who were registered in the fields of medicine, medical sciences, pharmacy, optometry, nursing sciences, and human nutrition and dietetics (HND).

A biostatistician from the university was consulted to estimate the sample size. The sample size was calculated using Slovin's formula.

$n = N / (1+Ne^2)$

n = study sample size; N = total number of registered students; and e = margin of error

$n = 1679/ (1+1679 \times 0.05^2)$

n = 322 students

A 10% attrition rate was added to the calculated sample size to cover for missing or incomplete data, resulting in a total sample of 354 students. The sample was proportionally distributed between various fields of study: Pharmacy (n = 61), Optometry (n = 40), Human Nutrition & Dietetics (HND) (n = 49), Medicine (n = 92), Medical Sciences (n = 52) and Nursing Sciences (n = 60) to achieve the desired sample.

## Data collection tools

All the data collection instruments, study information leaflets, and consent forms were available only in English, as the medium of instruction and communication at the study site is English. All the data collection instruments were self-administered.

**Socio-demographic and health-related questionnaire.** The socio-demographic and health-related information, such as gender, race, age, field of study, religious affiliation, household income, family residence, relationship status, history of psychiatric illness, whether they have taken any medicine for treating psychiatric illness, and who they are currently living with was obtained from the participants.

**Positive mental health instrument.** The researchers used a multi-dimensional PMH instrument developed by Vaingankar et al., (2011) [40]. The PMH instrument is a self-administered tool that covers all key and culturally appropriate domains of mental health and can be applied to compare levels of mental health across different populations. This instrument was developed through qualitative investigations of people with mental disorders, followed by quantitative and psychometric analysis. The research team obtained authorisation from the authors of the PMH instrument to use it in this study. This includes a copyright grant and the scoring algorithm.

This instrument has 57 questions, which include six domains:
The domains include:

1. **General coping** refers to individuals' responses and coping strategies during stressful situations and their ability to think positively and participate in selected activities.

2. **Emotional support** is key for helping people cope with difficult situations in life and for making them feel loved and wanted. A willingness to share the burden with others is important to obtain compassionate advice and care.

3. **Spirituality** encompasses both spiritual and religious practices and beliefs that influence individual's faith and behaviour in life. This contributes to PMH as a coping mechanism and means of establishing strong social support and networks.

4. **Interpersonal skills** are associated with all aspects of mental health and are essential in helping the individual develop and maintain good relationships, which in turn provide the support and network needed during times of distress.

5. **Personal Growth and Autonomy** mean knowing one's goals and ways to achieve them, which is a sign of good mental health. It reflects the level of confidence, freedom, sense of purpose, and the ability to self-evaluate and make decisions.

6. **Global Affect** is the experience of a positive mood, which is a sign of mental health. Calm, happiness, and enthusiasm mean emotional stability and are full of energy.

For the first five domains, participants were requested to mark how much each item describes them on a scale between 1 to 6 (1- 'Not at all like me' to 6-'Exactly like me'). For the 'Global affect' domain, they were requested to indicate 'how often over the past four weeks they felt calm, peaceful, relaxed, and enthusiastic' using a 5-point response scale (1- 'Never or very rarely' to 5- 'Very often or always'). Domain-specific scores were calculated by summing the scores of the respective items and dividing by the total number of items in each domain.

## Data collection

Data were collected over three weeks, from September 2023 to October 2023, by the research team. The research team designed a study information leaflet that provided detailed

information about the study, possible benefits, and implications. The researchers were trained on the data collection instruments, study information leaflet, and consent form to clarify and address the concerns raised by the participants. The students who met the inclusion criteria were conveniently enrolled in the study. The following inclusion criteria were applied: registered undergraduate students of FHS from the first to the final level, who are older than 18 years and who agreed to participate and sign the consent form. Students from the Department of Public Health (undergraduate programme not offered), postgraduate students, and students who declined to participate were excluded from the study.

On the day of data collection, the researchers briefed the participants about the study after the conclusion of their classes, explained the purpose and importance of this study, and requested their participation. The questionnaires were administered only after the participants had given written consent to participate in the study. The participants were given sufficient time to complete the questionnaires.

## Validity and reliability

Globally available instruments focused on specific domains of mental health in greater detail, using lengthy or too short questionnaires to provide significant comparisons and detection of changes across the domains. However, the PMH instrument utilised in this study comprehensively evaluated the multi-dimensional domains of mental health, which is appropriate for a South African study. The PMH instrument is a cross-cultural questionnaire developed by Vaingankar et al., (2011) and further validated by studies conducted in Singapore and South Africa [41, 42].

To ensure the validity of the data in this study, the researchers recruited the sample, which was representative of the target population, thus increasing the external validity and generalisability of the findings. Rigorous data collection methods were employed by the researchers in this study. Only the research team was involved in briefing the participants and during data collection to maintain consistency with data collection procedures. The researchers implemented data validation procedures by double-checking the thoroughness and completeness of the data. Data entry was verified for the correctness and completeness prior to data analysis to enhance reliability. The inclusion and exclusion criteria set for this study were strictly applied.

## Bias

Selection bias was minimised by strictly applying the inclusion and exclusion criteria. All the data collection instruments were self-administered, avoiding interviewer bias. The introduction of translation bias was excluded as all the questionnaires and consent forms were available only in English.

## Data analysis

A biostatistician from the university was consulted for assistance and advise on the data analysis. Responses from the questionnaires were exported to Microsoft Office Excel and analysed using the IBM Statistical Package for Social Studies (SPSS) version 29. The PMH scores were calculated using the scoring script provided by the authors of the PMH instrument. Recorded item scores for the questions specific to each domain were pooled together and divided by a number of questions to obtain the domain scores. The PMH scores ranged from 1 (lower PMH) to 6 (higher PMH). The statistical significance for the study was set at $p \leq 0.05$ using 2-sided tests. Independent t-tests and Analysis of Variance (ANOVA) were used to establish differences in mean scores between Total PMH and specific domains and socio-demographic and health-related variables.

## Ethical considerations

Ethical clearance (TREC/518/2023:UG) to conduct the study was obtained from the university's research ethics committee. Permission to conduct the study was obtained from the Registrar of the university. Authorisation to use the PMH instrument was obtained from the National Institute of Mental Health, Singapore.

A participant information leaflet was provided to the students, informing them of the objectives of the study, that participation in the study is voluntary, that they have the right to withdraw from the study at any time without providing reasons, and that their identity will remain anonymous. All the concerns of the prospective participants about the study were clarified before the commencement of data collection. All the participants have provided written consent before participating in the study. The study adhered to all ethical standards and ethical research practices, following the Declaration of Helsinki. All the records and data were kept in a safe and secure place to maintain confidentiality, with access available only to the research team.

## Results

The study population comprised 354 FHS students from the first year to the final year with a mean age of 20.6 years. Table 1 displays the socio-demographic and health-related characteristics of the respondents. The majority of the students were black (99.2%, n = 351), single (72%, n = 255), received study bursaries from the government (78.5%, n = 278), hail from a rural area (77.7%, n = 275) and resided in the university campus (74.5%, n = 246). More than half of the students were female (59.9%), affiliated with Christianity (69.2%), and had a household income of less than 10 000 South African Rand (ZAR) per month (50.4%, n = 182). Most of the students indicated that they did not have a history of any psychiatric illness (96%, n = 340) or had taken any psychiatric medicines before (96.3%, n = 341).

## Total PMH and domain-specific scores by socio-demographic and health-related variables

Table 2 displays the mean total PMH and domain-specific scores by socio-demographic and health-related variables. The total PMH scores ranged from 4.24 to 4.97 suggesting that the students of FHS reported a moderate to higher level of PMH. Significant differences were observed in the total PMH and domain mean scores across various socio-demographic and health-related variables. Gender ($p = 0.037$), age ($p = 0.043$) and field of study ($p = 0.016$) had a significant influence on total PMH score. In the general coping domain, significant differences in mean scores were observed with gender ($p < 0.001$), where males reporting significantly higher scores as compared to females. Significant differences in mean scores regarding the emotional support domain were observed with race ($p = 0.039$) and relationship status ($p = 0.005$). There is a significant difference in mean scores between the field of study ($p = 0.043$), gender ($p = 0.008$) and religious affiliation ($p = 0.009$) on spirituality. In the case of the interpersonal skills domain, a significant difference in mean scores was observed between the field of study ($p = 0.025$) and current living status ($p = 0.008$). With respect to personal growth of autonomy, significant differences in mean scores were observed in age ($p = 0.022$), field of study ($p = 0.002$), history of psychiatric illness ($p = 0.002$) and gender ($p = 0.003$). In the global affect, a significant difference in mean score was observed with age ($p < 0.001$) and gender ($p < 0.001$).

## Discussion

It is evident from the study findings that most of the students that participated in this study were black, from a rural background, with low household income, and stayed at the university

**Table 1. Socio-demographic characteristics of the sample (n = 354).**

| Socio-demographic and health-related characteristics | | Number of students (n = 354) | Percentage (%) |
|---|---|---|---|
| Gender | Male | 142 | 40.1 |
| | Female | 212 | 59.9 |
| Race | Black | 351 | 99.2 |
| | White | 3 | 0.8 |
| Age (in years) | 18–19 | 96 | 27.2 |
| | 20–21 | 131 | 37.0 |
| | 22–23 | 84 | 23.7 |
| | >23 | 43 | 12.1 |
| Degree | Pharmacy | 61 | 17.2 |
| | Optometry | 40 | 11.3 |
| | Human Nutrition & Dietetics | 49 | 13.8 |
| | Medicine | 92 | 26.1 |
| | Medical Sciences | 52 | 14.7 |
| | Nursing | 60 | 16.9 |
| Bursary | Yes | 278 | 78.5 |
| | No | 76 | 21.5 |
| Religious affiliation | Christian | 245 | 69.2 |
| | Other | 109 | 30.8 |
| Household income (ZAR per month) | 0–5000 | 103 | 29.1 |
| | 5001–10000 | 79 | 22.3 |
| | 10001–20000 | 73 | 21.5 |
| | > 20000 | 96 | 27.1 |
| Family residence | Rural | 275 | 77.7 |
| | Urban | 79 | 22.3 |
| Current living status | In university residence | 246 | 74.5 |
| | Private residence | 82 | 23.2 |
| | Living with parents | 8 | 2.3 |
| Relationship status | Single | 255 | 72.0 |
| | In a relationship/Married | 99 | 28.0 |
| History of psychiatric illness | Yes | 14 | 4.0 |
| | No | 340 | 96.0 |
| Have you taken any psychiatric medicines before? | Yes | 13 | 3.7 |
| | No | 341 | 96.3 |

residence. This can be attributed to the location of the university, which is in the rural and poorest part of South Africa with a predominantly black population. The university was established during apartheid to provide education to blacks [43, 44]. The university continues to be a first choice for students from the disadvantaged communities in the province due to its proximity to their homes [45].

Our findings showed significant differences in the levels of the total PMH between men and women. However, most of the students with a low level of PMH were female, which aligns with the findings of previous research [46, 47]. For college students, stress is a major problem, and they regard their college years as among the most stressful times of their lives. Male students have better coping skills with stress as they become more proactive in their stress response. It has also been demonstrated that female students experience higher levels of general and academic stress than their male counterparts [48] which could have influenced their total PMH levels. However, other studies did not identify any total perceived stress differences

**Table 2. Total PMH and domain scores by socio-demographic and health-related characteristics.**

| Socio-demographic and health-related characteristics | | Total PMH | | General coping | | Emotional support | | Spirituality | | Interpersonal skills | | Personal growth and autonomy | | Global affect | |
|---|---|---|---|---|---|---|---|---|---|---|---|---|---|---|---|
| | | Mean (±SD) | P | Mean (±SD) | P | Mean (±SD) | P | Mean (±SD) | P | Mean (±SD) | p | Mean (±SD) | p | Mean (±SD) | p |
| **Gender** | Male | 4.50 (0.68) | 0.037* | 4.29 (0.91) | <0.001* | 4.36 (1.12) | 0.251 | 4.73 (1.03) | 0.008*] | 4.33 (0.87) | 0.145 | 4.78 (0.96) | 0.003* | 4.55 (1.06) | <0.001 |
| | Female | 4.35 (0.61) | | 3.88 (1.03) | | 4.51 (1.25) | | 5.01 (0.95) | | 4.20 (0.83) | | 4.50 (0.82) | | 4.10 (1.15) | |
| **Race** | Black | 4.41 (0.64) | 0.132 | 4.04 (1.00) | 0.494 | 4.44 (1.19) | 0.039* | 4.90 (0.99) | 0.775 | 4.25 (0.85) | 0.184 | 4.61 (0.89) | 0.122 | 4.28 (1.14) | 0.894 |
| | White | 4.97 (0.40) | | 4.43 (0.29) | | 5.87 (0.15) | | 4.73 (1.48) | | 4.90 (0.95) | | 5.40 (0.66) | | 4.37 (0.51) | |
| **Bursary** | Yes | 4.42 (0.65) | 0.566 | 4.05 (1.00) | 0.688 | 4.47 (1.23) | 0.574 | 4.91 (0.98) | 0.615 | 4.26 (0.87) | 0.890 | 4.62 (0.88) | 0.793 | 4.30 (1.13) | 0.543 |
| | No | 4.38 (0.60) | | 4.00 (0.99) | | 4.38 (1.07) | | 4.85 (1.05) | | 4.24 (0.79) | | 4.59 (0.92) | | 4.21 (1.16) | |
| **Relationship status** | Single | 4.39 (0.67) | 0.379 | 4.04 (1.05) | 0.951 | 4.34 (1.26) | 0.005* | 4.92 (0.96) | 0.398 | 4.27 (0.89) | 0.617 | 4.59 (0.91) | 0.414 | 4.23 (1.14) | 0.209 |
| | In a relationship/ Married | 4.46 (0.56) | | 4.03 (0.85) | | 4.74 (0.96) | | 4.83 (1.07) | | 4.22 (0.74) | | 4.67 (0.82) | | 4.40 (1.10) | |
| **Age (in years)** | 18–19 | 4.28 (0.65) | 0.043* | 3.90 (1.02) | 0.158 | 4.32 (1.25) | 0.163 | 4.87 (0.96) | 0.975 | 4.27 (0.81) | 0.989 | 4.41 (0.91) | 0.022* | 3.91 (1.28) | <0.001* |
| | 20–21 | 4.44 (0.66) | | 4.08 (1.04) | | 4.38 (1.27) | | 4.92 (1.04) | | 4.24 (0.98) | | 4.68 (0.87) | | 4.43 (1.02) | |
| | 22–23 | 4.41 (0.54) | | 4.00 (0.93) | | 4.56 (1.02) | | 4.88 (0.95) | | 4.24 (0.71) | | 4.60 (0.85) | | 4.29 (1.05) | |
| | >23 | 4.60 (0.71) | | 4.30 (0.95) | | 4.74 (1.13) | | 4.93 (1.04) | | 4.28 (0.78) | | 4.87 (0.90) | | 4.63 (1.07) | |
| **Degree** | Pharmacy | 4.28 (0.64) | 0.016* | 3.97 (1.03) | 0.864 | 4.19 (1.24) | 0.115 | 4.92 (0.77) | 0.043* | 4.17 (0.76) | 0.025* | 4.40 (0.85) | 0.002* | 4.06 (1.02) | 0.240 |
| | Optometry | 4.24 (0.72) | | 3.94 (1.12) | | 4.40 (1.13) | | 4.77 (0.97) | | 4.03 (0.82) | | 4.26 (1.08) | | 4.21 (1.14) | |
| | Nursing | 4.43 (0.62) | | 4.17 (1.04) | | 4.60 (1.23) | | 4.61 (1.17) | | 4.20 (0.80) | | 4.72 (0.79) | | 4.24 (1.20) | |
| | HND | 4.59 (0.63) | | 4.01 (1.02) | | 4.46 (1.16) | | 5.22 (0.81) | | 4.57 (1.17) | | 4.81 (0.87) | | 4.51 (1.23) | |
| | Medicine | 4.53 (0.57) | | 4.08 (0.95) | | 4.68 (1.16) | | 4.97 (0.95) | | 4.34 (0.69) | | 4.80 (0.83) | | 4.43 (1.14) | |
| | Medical science | 4.31 (0.67) | | 4.01 (0.92) | | 4.24 (1.19) | | 4.87 (1.18) | | 4.12 (0.85) | | 4.49 (0.86) | | 4.16 (1.03) | |
| **Household income (in ZAR)** | 0–5000 | 4.34 (0.68) | 0.472 | 3.92 (1.04) | 0.349 | 4.30 (1.23) | 0.137 | 4.90 (0.95) | 0.985 | 4.30 (1.02) | 0.484 | 4.56 (0.97) | 0.794 | 4.13 (1.23) | 0.287 |
| | 5001–10000 | 4.48 (0.62) | | 4.16 (0.88) | | 4.71 (1.12) | | 4.91 (1.02) | | 4.27 (0.70) | | 4.66 (0.87) | | 4.23 (1.21) | |
| | 10001–20000 | 4.46 (0.61) | | 4.12 (0.98) | | 4.43 (1.26) | | 4.93 (1.00) | | 4.31 (0.84) | | 4.67 (0.78) | | 4.34 (1.05) | |
| | > 20000 | 4.39 (0.64) | | 4.00 (1.05) | | 4.42 (1.14) | | 4.87 (1.04) | | 4.14 (0.76) | | 4.58 (0.90) | | 4.43 (1.01) | |
| **Family residence** | Rural | 4.41 (0.66) | 0.997 | 4.03 (1.01) | 0.612 | 4.43 (1.22) | 0.439 | 4.92 (0.96) | 0.372 | 4.28 (0.87) | 0.317 | 4.62 (0.89) | 0.859 | 4.27 (1.14) | 0.700 |
| | Urban | 4.41 (0.59) | | 4.09 (0.97) | | 4.54 (1.08) | | 4.81 (1.10) | | 4.17 (0.75) | | 4.60 (0.89) | | 4.32 (1.11) | |

(*Continued*)

**Table 2.** (Continued)

| Socio-demographic and health-related characteristics | | Total PMH | | General coping | | Emotional support | | Spirituality | | Interpersonal skills | | Personal growth and autonomy | | Global affect | |
|---|---|---|---|---|---|---|---|---|---|---|---|---|---|---|---|---|
| | | Mean (±SD) | P | Mean (±SD) | P | Mean (±SD) | P | Mean (±SD) | P | Mean (±SD) | p | Mean (±SD) | p | Mean (±SD) | p |
| **Religious affiliation** | Christianity | 4.43 (0.64) | 0.364 | 4.04 (0.99) | 0.898 | 4.46 (1.19) | 0.823 | 4.99 (0.94) | 0.009* | 4.28 (087) | 0.315 | 4.61 (0.87) | 0.958 | 4.27 (1.15) | 0.812 |
| | None | 4.37 (0.65) | | 4.03 (1.03) | | 4.43 (1.21) | | 4.69 (1.07) | | 4.18 (0.79) | | 4.62 (0.92) | | 4.30 (1.11) | |
| **Current living status** | In university residence | 4.40 (0.64) | 0.905 | 4.04 (1.01) | 0.823 | 4.47 (1.20) | 0.811 | 4.87 (1.00) | 0.633 | 4.17 (0.79) | 0.008* | 4.62 (0.91) | 0.650 | 4.33 (1.13) | 0.180 |
| | Private residence | 4.44 (0.66) | | 4.06 (1.00) | | 4.38 (1.23) | | 4.98 (0.95) | | 4.50 (0.99) | | 4.55 (0.85) | | 4.16 (1.12) | |
| | Living with parents | 4.40 (0.64) | | 3.83 (0.68) | | 4.56 (0.77) | | 5.04 (1.10) | | 4.31 (0.90) | | 4.81 (0.75) | | 3.73 (1.30) | |
| **History of psychiatric illness** | Yes | 4.08 (0.70) | 0.054 | 3.77 (0.86) | 0.321 | 3.89 (1.27) | 0.080 | 4.97 (1.11) | 0.789 | 4.26 (0.77) | 0.965 | 3.87 (1.15) | 0.002* | 3.80 (1.15) | 0.120 |
| | No | 4.43 (0.64) | | 4.05 (1.00) | | 4.47 (1.19) | | 4.89 (0.99) | | 4.25 (0.85) | | 4.64 (0.87) | | 4.30 (1.13) | |
| **Have you taken psychiatric medicine before?** | Yes | 4.16 (0.66) | 0.128 | 3.71 (0.84) | 0.205 | 4.10 (1.34) | 0.260 | 4.71 (1.19) | 0.466 | 4.23 (0.71) | 0.918 | 4.18 (1.11) | 0.062 | 4.12 (0.96) | 0.595 |
| | No | 4.42 (0.64) | | 4.05 (1.00) | | 4.47 (1.19) | | 4.90 (0.98) | | 4.25 (0.85) | | 4.63 (0.87) | | 4.29 (1.14) | |

*$p \leq 0.05$; SD: Standard Deviation; HND: Human Nutrition & Dietetics

in their college populations, which contrasts with our study findings [49, 50]. As illustrated, the existing research has produced conflicting results about the relationship between gender and perceived stress levels, which calls for further investigation. The possible explanation for significantly low levels of total PMH in female students could also be linked to menstruation, as there was an association between menstrual-related symptoms and the levels of psychological distress [51]. The above finding highlights the need for South African universities to implement gender-specific interventional programmes that address gender-specific risk factors.

Age was found to be significantly associated with total PMH, with students between the ages of 18 and 19 years (first year of study) reporting significantly lower mean scores compared to other age groups in most domains, excluding emotional support and interpersonal skills. Our study finding is supported by studies conducted by WHO and Dessie et al. (2013) in Ethiopia, which found that students in the first year of study experienced a greater variety of stressors than any other group of students [25, 52]. The finding may be attributed to the fact that first-year students are in their transition to university life, and the added pressure of heightened social and academic expectations puts them at risk for mental health issues. In addition, first year students' inability to navigate their way and cope with the stress of such a transition, loneliness, and homesickness could have significantly influenced their PMH levels [34, 53]. Results from a study conducted among Malaysian university students revealed that factors associated with financial difficulties, demands of the university environment and the university's administrative processes, and non-academic related issues were reported as underlying factors in first-year students' low levels of mental well-being [54]. A study conducted by Mason (2019) with a sample of 55 first-year university students in South Africa highlighted the need for more South African research on the application of positive psychology to assist students in navigating the stressful first-year experience by identifying, developing, and applying signature strengths [55].

In this study, the field of study was found to be significantly associated with total PMH, with students in HND reporting higher mean scores in most domains except general coping and emotional support when compared to the students from other fields of study. Although there is no specific reason to link their significantly higher levels of total PMH, the authors assume that their curriculum is less intense as compared to other undergraduate degrees offered by FHS.

This study identified a significant association between relationship status and emotional support. Students who were either in a relationship or married reported significantly higher scores compared to students who were single in the emotional support domain. Over the course of a person's life, being single consistently correlates with poor mental health, in contrast to marriage. Literature further suggests that, compared to marriage, being single is one of the risk factors for depressive symptoms in both men and women. Our study finding is supported by existing literature that being single is associated with poorer levels of PMH in younger people, indicating that living circumstances or legal status are not as significant at this age [56, 57].

## Limitations

This is a cross-sectional study conducted with a comparatively smaller group of students in FHS at a rural university in South Africa. Therefore, we cannot generalise the results to the entire student population in South Africa. Therefore, this study may not have established causal relationships between variables and may have only provided information on associations.

The authors acknowledge the possibility of response bias in this study as some students may have biases or motivations that influence their responses, which may have led to inaccurate or skewed data. As this study was conducted at a specific institution and in a geographical area of South Africa that has a predominant black population, it excluded students from other races. Hence, the researchers cannot justify the effect of race on total PMH and other domains of PMH. Despite the fact that students with a history of psychiatric illness and those receiving treatment reported lower levels of PMH, the limited sample makes the findings unjustifiable. Another limitation of this study is social desirability, as some students may have responded in a socially desirable way, which may have yielded inaccurate or biassed responses.

Although these limitations exist, the researchers believe that the results of this study are robust, convincing, and provide a platform for future research to further investigate the PMH levels in various university students with a larger study population and compare the levels of PMH in urban and rural universities in South Africa.

## Conclusion

This is the first study in South Africa to measure the levels of PMH among the FHS students at a rural university in South Africa using a multi-dimensional PMH instrument. The study findings clearly suggest that there is a place for gender-specific interventional programmes addressing gender-specific risk factors in South African universities. Interventions must be aimed at providing a platform for educating students regarding issues, such as mental distress, that are typically encountered during the first year of study, and the importance of developing a range of effective social supports. Higher education institutions should make student mental health and well-being a priority issue, as it directly influences the learning process and adjustment to the academic environment. By creating a culture of well-being and providing resources and support services, universities can contribute to positive mental health outcomes for their students.

## Supporting information

**S1 Checklist. CROSS checklist.**
(DOCX)

**S1 Data.**
(DOCX)

**S1 Table.**
(XLSX)

**S2 Table.**
(XLSX)

## Author Contributions

**Conceptualization:** Rajesh Vikram Vagiri.

**Data curation:** Rajesh Vikram Vagiri.

**Formal analysis:** Rajesh Vikram Vagiri, Phuty Elizabeth Leboho, Lokwene Katlego Desry.

**Investigation:** Rajesh Vikram Vagiri, Phuty Elizabeth Leboho, Lokwene Katlego Desry, Machaka Khutso, Mbedzi Pfunzo.

**Methodology:** Rajesh Vikram Vagiri, Lokwene Katlego Desry.

**Project administration:** Rajesh Vikram Vagiri, Phuty Elizabeth Leboho, Mbedzi Pfunzo.

**Supervision:** Rajesh Vikram Vagiri.

**Writing – original draft:** Rajesh Vikram Vagiri, Phuty Elizabeth Leboho, Lokwene Katlego Desry, Machaka Khutso, Mbedzi Pfunzo.

**Writing – review & editing:** Rajesh Vikram Vagiri, Phuty Elizabeth Leboho, Lokwene Katlego Desry, Machaka Khutso, Mbedzi Pfunzo.

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
