## [Decision Letter · Decision Letter 0]

23 Nov 2023

PGPH-D-23-02265

Investigation of positive mental health levels among faculty of health sciences students at a rural university in South Africa

Dear Dr. Vagiri,

Thank you for submitting your manuscript to PLOS Global Public Health. After careful consideration, we feel that it has merit but does not fully meet PLOS Global Public Health’s publication criteria as it currently stands. Therefore, we invite you to submit a revised version of the manuscript that addresses the points raised during the review process.

Thank you for this very interesting and relevant paper. Both the authors recognized the need for this topic. I believe your paper fulfils the publication criterion of the journal. Please address the comments raised by the reviewers especially those with regards to expanding the literature review and providing more information about the administration of the tool.

We look forward to receiving your revised manuscript.

Kind regards,

Susmita Chandramouleeshwaran

Academic Editor

Journal Requirements:

1. We do not publish any copyright or trademark symbols that usually accompany proprietary names, eg  ©, ®, ™  (e.g. next to drug or reagent names). Please remove all instances of trademark/copyright symbols throughout the text, including ® on pages 1 and 6.

Additional Editor Comments (if provided):

Reviewers' comments:

Reviewer's Responses to Questions

**Comments to the Author**

1. Does this manuscript meet PLOS Global Public Health’s publication criteria? Is the manuscript technically sound, and do the data support the conclusions? The manuscript must describe methodologically and ethically rigorous research with conclusions that are appropriately drawn based on the data presented.

Reviewer #1: Yes

Reviewer #2: Partly

2. Has the statistical analysis been performed appropriately and rigorously?

Reviewer #1: Yes

Reviewer #2: No

3. Have the authors made all data underlying the findings in their manuscript fully available (please refer to the Data Availability Statement at the start of the manuscript PDF file)?

Reviewer #1: Yes

Reviewer #2: Yes

4. Is the manuscript presented in an intelligible fashion and written in standard English?

Reviewer #1: Yes

Reviewer #2: Yes

5. Review Comments to the Author

Reviewer #1: A well-conducted, survey, valid and reliable results. Encourage the authors to consider the following to improve the manuscript, keeping in mind the global audience who would possibly read this well-conducted research with the following questions in mind:

1. What is the current breadth, width and depth of positive psychology in South Africa? As the authors might be aware, literature in positive mental health has a large representation in the field of positive psychology and having a section on current best research on positive psychology in students in South Africa will help. Some references in this regard are (non-exhaustive list:) https://files.eric.ed.gov/fulltext/EJ1237700.pdf ; https://journals.sagepub.com/doi/10.1177/008124630703700307;
https://link.springer.com/book/10.1007/978-94-007-6368-5 ; https://www.ncbi.nlm.nih.gov/pmc/articles/PMC9705334/.

2. Are there known/unknown differences between positive mental health in urban and rural communities? Authors will benefit from adding a section on this in Introduction. Interesting references (non-exhaustive) are: https://psycnet.apa.org/record/2013-17535-020 ; https://www.jstor.org/stable/41409358

3. please consider revising line 58 in introduction: " A PMH for students is very important as it affects their ability to learn, achieve academically, and behave appropriately.". Suggest to limit to the previous objective evidence of existing correlations between positive mental health and outcomes.

4. paragraph with lines 56-64: point well made, but the relationship between positive mental health and mental illness is much more complex than just presence of one=absence/limiting the other. refer to the two continua model here: Ref: Keyes CM. The Mental Health Continuum: From languishing to flourishing in life. Journal of Health and Social Research 2002; 43(June): 207-22.

5. Suggest revising lines 69-72 as there is some research, as pointed above in point 1, in this area in South Africa.

6. Data collection and data analysis: Specifics about who collected the data and who analysis it would help improve robustness of the study. Please consider adhering to the checklist of reporting of survey studies (CROSS) developed by the EQUATOR framework and adding the details in the text accordingly as well as providing the completed checklist to the readers of this journal.

7. Consider revising line 246-247. The gender differences may be much more complex than attributed to hormonal changes in menstruation, including role differences in specific cultures etc. Also, the reference 30, to the best of my knowledge, does not allude to hormonal changes in menstruation as a major factor.

8. Regarding validity of PMH instrument, consider adding details of language of communication/administration of the scale (whether English or local language) and whether that was a limitation or not.

9. limitations section: please consider including limitations of survey design itself that specifically apply to this research, for example, sampling error and non-response error, as well as the influence of question wording and format. Additionally, survey results can be affected by respondent motivation, availability, and willingness.

Thank you for allowing me to review this work.

I am happy to review any questions or clarifications with the authors/editors.

Thank you,

Dr. Dholakia

Reviewer #2: Dear Prof Rajesh Vagiri,

You have chosen a very interesting topic. Mental health is seen more in pathological terms. You endeavored to work from positive mental health perspective. However I think there are certain flaws in your analysis. I have pointed them out in the attached the attached document. You have a lot of scope to improve the article and add to understanding of positive mental health.I have mentioned that in the document.

Best wishes

Reviewer

6. PLOS authors have the option to publish the peer review history of their article (what does this mean?). If published, this will include your full peer review and any attached files.

**Do you want your identity to be public for this peer review?** For information about this choice, including consent withdrawal, please see our Privacy Policy.

Reviewer #1: **Yes: **saumil dholakia

Reviewer #2: **Yes: **Dr Dheeraj Kattula

---

## [Decision Letter · Decision Letter 1]

26 Apr 2024

PGPH-D-23-02265R1

Investigation of positive mental health levels among faculty of health sciences students at a rural university in South Africa

Dear Dr. Vagiri,

Thank you for submitting your manuscript to PLOS Global Public Health. Before we can recommend acceptance, we request that you kindly address the reviewers' outstanding queries. Therefore, we invite you to submit a revised version of the manuscript that addresses the points raised during the review process.

Your manuscript has been assessed by 4 reviewers and their comments are available below. Could you please revise your manuscript carefully to address all their outstanding comments?

We look forward to receiving your revised manuscript.

Kind regards,

Annesha Sil, Ph.D.

PLOS Staff Editor 

Journal Requirements:

Additional Editor Comments (if provided):

Reviewers' comments:

Reviewer's Responses to Questions

**Comments to the Author**

1. If the authors have adequately addressed your comments raised in a previous round of review and you feel that this manuscript is now acceptable for publication, you may indicate that here to bypass the “Comments to the Author” section, enter your conflict of interest statement in the “Confidential to Editor” section, and submit your "Accept" recommendation.

Reviewer #1: All comments have been addressed

Reviewer #2: All comments have been addressed

Reviewer #3: All comments have been addressed

Reviewer #4: (No Response)

2. Does this manuscript meet PLOS Global Public Health’s publication criteria? Is the manuscript technically sound, and do the data support the conclusions? The manuscript must describe methodologically and ethically rigorous research with conclusions that are appropriately drawn based on the data presented.

Reviewer #1: Yes

Reviewer #2: Yes

Reviewer #3: Yes

Reviewer #4: Yes

3. Has the statistical analysis been performed appropriately and rigorously?

Reviewer #1: Yes

Reviewer #2: No

Reviewer #3: Yes

Reviewer #4: Yes

4. Have the authors made all data underlying the findings in their manuscript fully available (please refer to the Data Availability Statement at the start of the manuscript PDF file)?

Reviewer #1: Yes

Reviewer #2: Yes

Reviewer #3: Yes

Reviewer #4: Yes

5. Is the manuscript presented in an intelligible fashion and written in standard English?

Reviewer #1: Yes

Reviewer #2: Yes

Reviewer #3: Yes

Reviewer #4: Yes

6. Review Comments to the Author

Reviewer #1: Much improved version, congratulate the authors to have dived in deep.

The manuscript can be reviewed for flow and grammar but otherwise meets the publication criteria and readers of this journal will benefit from this well-conducted survey on positive mental health in South Africa.

Reviewer #2: Dear authors,

Table 3 is wrong. You cannot do correlations between categorical variable like gender, race etc with continuous variables from scale scores. Kindly consult a statistics staff and do the needful.

You are to compare variables like total PMH between men and women using t test, degree and PMH using ANOVA/Kruskal Wallis test depending on distribution

Please do the needful and resubmit.

I am happy about other changes.

Reviewer #3: Authors have confused the the terms "affect" and "effect" in many places.

Reviewer #4: (No Response)

7. PLOS authors have the option to publish the peer review history of their article (what does this mean?). If published, this will include your full peer review and any attached files.

**Do you want your identity to be public for this peer review?** For information about this choice, including consent withdrawal, please see our Privacy Policy.

Reviewer #1: No

Reviewer #2: No

Reviewer #3: No

Reviewer #4: **Yes: **Dr.Thanabalasingam Gadambanathan

---

## [Decision Letter · Decision Letter 2]

15 May 2024

PGPH-D-23-02265R2

Investigation of positive mental health levels among faculty of health sciences students at a rural university in South Africa

Dear Dr. Vagiri,

Thank you for submitting your manuscript to PLOS Global Public Health. After careful consideration, we feel that it has merit but does not fully meet PLOS Global Public Health’s publication criteria as it currently stands. Therefore, we invite you to submit a revised version of the manuscript that addresses the points raised during the review process.

We look forward to receiving your revised manuscript.

Kind regards,

Medhin Selamu Tegegn

Academic Editor

Journal Requirements:

Additional Editor Comments (if provided):

Reviewers' comments:

Reviewer's Responses to Questions

**Comments to the Author**

1. If the authors have adequately addressed your comments raised in a previous round of review and you feel that this manuscript is now acceptable for publication, you may indicate that here to bypass the “Comments to the Author” section, enter your conflict of interest statement in the “Confidential to Editor” section, and submit your "Accept" recommendation.

Reviewer #4: All comments have been addressed

2. Does this manuscript meet PLOS Global Public Health’s publication criteria? Is the manuscript technically sound, and do the data support the conclusions? The manuscript must describe methodologically and ethically rigorous research with conclusions that are appropriately drawn based on the data presented.

Reviewer #4: Yes

3. Has the statistical analysis been performed appropriately and rigorously?

Reviewer #4: Yes

4. Have the authors made all data underlying the findings in their manuscript fully available (please refer to the Data Availability Statement at the start of the manuscript PDF file)?

Reviewer #4: Yes

5. Is the manuscript presented in an intelligible fashion and written in standard English?

Reviewer #4: Yes

6. Review Comments to the Author

Reviewer #4: No comments or suggestions

7. PLOS authors have the option to publish the peer review history of their article (what does this mean?). If published, this will include your full peer review and any attached files.

**Do you want your identity to be public for this peer review?** For information about this choice, including consent withdrawal, please see our Privacy Policy.

Reviewer #4: **Yes: **Thanabalasingam Gadambanathan

---

## [Editor Report · Decision Letter 3]

31 May 2024

PGPH-D-23-02265R3

Investigation of positive mental health levels among faculty of health sciences students at a rural university in South Africa

Dear Dr. Vagiri,

Thank you for submitting your manuscript to PLOS Global Public Health. After careful consideration, we feel that it has merit but does not fully meet PLOS Global Public Health’s publication criteria as it currently stands. Therefore, we invite you to submit a revised version of the manuscript that addresses the points raised during the review process.

We look forward to receiving your revised manuscript.

Kind regards,

Medhin Selamu Tegegn

Academic Editor
---

## [Editor Report · Decision Letter 4]

5 Jul 2024

Investigation of positive mental health levels among faculty of health sciences students at a rural university in South Africa

PGPH-D-23-02265R4

Dear Dr Vagiri,

We are pleased to inform you that your manuscript 'Investigation of positive mental health levels among faculty of health sciences students at a rural university in South Africa' has been provisionally accepted for publication in PLOS Global Public Health.

Best regards,

Medhin Selamu Tegegn

Academic Editor